# Measurement of airborne particle emission during surgical and percutaneous dilatational tracheostomy COVID-19 adapted procedures in a swine model: Experimental report and review of literature

**Valentin Favier**[1,2]*, **Mickael Lescroart**[3], **Benjamin Pequignot**[3], **Léonie Grimmer**[4],
**Arnaud Florentin**[4,5], **Patrice Gallet**[6,7,8]

1 Department of Otolaryngology-Head and Neck Surgery, Gui de Chauliac Hospital, University Hospital of Montpellier, Montpellier, France, 2 Montpellier Laboratory of Informatics, Robotics and Microelectronics (LIRMM), ICAR Team, French National Centre for Scientific Research (CNRS), Montpellier University, Montpellier, France, 3 Intensive Care Unit Brabois, University Regional Hospital of Nancy, Vandoeuvre-lès-Nancy, France, 4 Department of Hygiene, Environmental Risks and Healthcare Associated Risks, University of Lorraine, Vandoeuvre-lès-Nancy, France, 5 Infection Prevention and Control Team, Regional University Hospital of Nancy, Vandœuvre-lès-Nancy, France, 6 ENT Department, Regional University Hospital of Nancy, University of Lorraine, Vandœuvre-lès-Nancy, France, 7 Virtual Hospital of Lorraine, University of Lorraine, Vandoeuvre-lès-Nancy, France, 8 NGERE, INSERM U1256 Lab, University of Lorraine, Vandoeuvre-lès-Nancy, France

* Valentin_favier@hotmail.com

## Abstract

### Introduction

Surgical tracheostomy (ST) and Percutaneous dilatational tracheostomy (PDT) are classified as high-risk aerosol-generating procedures and might lead to healthcare workers (HCW) infection. Albeit the COVID-19 strain slightly released since the vaccination era, preventing HCW from infection remains a major economical and medical concern. To date, there is no study monitoring particle emissions during ST and PDT in a clinical setting. The aim of this study was to monitor particle emissions during ST and PDT in a swine model.

### Methods

A randomized animal study on swine model with induced acute respiratory distress syndrome (ARDS) was conducted. A dedicated room with controlled airflow was used to standardize the measurements obtained using an airborne optical particle counter. 6 ST and 6 PDT were performed in 12 pigs. Airborne particles (diameter of 0.5 to 3 µm) were continuously measured; video and audio data were recorded. The emission of particles was considered as significant if the number of particles increased beyond the normal variations of baseline particle contamination determinations in the room. These significant emissions were interpreted in the light of video and audio recordings. Duration of procedures, number of expiratory pauses, technical errors and adverse events were also analyzed.

**Data Availability Statement:** All relevant data are within the manuscript and its Supporting Information files.

**Funding:** VF received funding to support a PhD thesis on simulation from November 2019 to November 2020 by the "Collège Français d'ORL et chirurgie cervico-faciale" and the Rotary International Club of Montpellier, France. The sponsors played no role in study design, data collection and analysis, decision to publish, or preparation of the manuscript. All other authors have no conflict of interest to disclose. The funders had no role in study design, data collection and analysis, decision to publish, or preparation of the manuscript.

**Competing interests:** The authors have declared that no competing interests exist.

## Results

10 procedures (5 ST and 5 PDT) were fully analyzable. There was no systematic aerosolization during procedures. However, in 1/5 ST and 4/5 PDT, minor leaks and some adverse events (cuff perforation in 1 ST and 1 PDT) occurred. Human factors were responsible for 1 aerosolization during 1 PDT procedure. ST duration was significantly shorter than PDT (8.6 ± 1.3 vs 15.6 ± 1.9 minutes) and required less expiratory pauses (1 vs 6.8 ± 1.2).

## Conclusions

COVID-19 adaptations allow preventing for major aerosol leaks for both ST and PDT, contributing to preserving healthcare workers during COVID-19 outbreak, but failed to achieve a perfectly airtight procedure. However, with COVID-19 adaptations, PDT required more expiratory pauses and more time than ST. Human factors and adverse events may lead to aerosolization and might be more frequent in PDT.

## 1 Introduction

The COVID-19 outbreak has been challenging so far for the medical field. It led to an unprecedented number of patients requiring prolonged mechanical ventilation and therefore tracheostomies. Tracheostomies, as well as endotracheal intubation, are classified as aerosol-generating procedures [1–3], exposing healthcare workers (HCWs) to a risk of viral infection [4]. Based on previous experiences (i.e., SARS-Cov-1 [5,6]), simulation studies and expert consensus, various adaptations have been proposed for both surgical [7–13] and percutaneous [7,14–23] tracheostomies, including suggestions of protective equipment [13,24] (Table 1). Despite many recommendations quickly issued to ensure the safety of HCWs performing tracheostomies [25,26], the safety differences between surgical tracheostomies (ST) and percutaneous dilatational tracheostomies (PDT) techniques are unknown [27]. During the Sars-CoV-1 outbreak, surgical tracheostomies were generally favored over percutaneous tracheostomies. Indeed, PDT is associated with different steps that might generate aerosol leaks (bronchoscopy,

**Table 1. Technical adaptation of surgical (ST) and percutaneous dilatational (PDT) tracheostomy procedures to minimize aerosolization.**

| | Covid-19 technical adaptations proposed in the literature |
|---|---|
| Both procedures | • Negative pressure ICU/operative room [7–11,16–18] or use of negative pressure enclosure [13,24]<br>• Patient fully paralyzed to minimize cough reflex [7–9,14–17,19,22]<br>• Pushing the tube as caudally as possible and ensure hyperinflation of the tube cuff [9–11,14–16,22–23]<br>• Adequate pre-oxygenation of the patient to optimize ventilation pauses [9,10,13,16,17,19]<br>• Expert physicians required [7,14,18,20–22] |
| ST | • Electrocautery avoidance [7–9]<br>• A single expiratory pause from tracheal entry to tracheostomy tube cuff inflation [7–12]<br>• Reducing the use of suction [8]<br>• Use of non-fenestrated cuffed tracheostomy tube [9]<br>• Interruption of ventilation 30 seconds before tracheal incision [10]<br>• Make the smallest incision possible [11]<br>• Use of a vertical incision below the cricoid level [11] or classical horizontal incision [12] |
| PDT | • Use of bronchoscopy [7,14–17,22,23]<br>• Expiratory pauses when placing the guidewire, performing dilatation and intratracheal placement of tracheostomy tube [7,14–17,22,23] |

dilatation steps), while ST is thought to lead to a unique potential exposure to aerosols when entering the trachea. In a recent attempt to summarize existing evidence and provide guidance for both healthcare providers and systems, it was stated that "the likelihood of aerosol generation is increased with percutaneous tracheostomy compared with surgical approaches" [28]. However, this statement does not rely on any experimental result. Therefore, the authors recommended to "continue to do tracheostomies using the techniques and equipment with which [teams] are familiar, and confident and experienced in using." In the USA and UK guidelines [3,29], either open surgical tracheostomy (ST) or percutaneous dilatational tracheostomy (PDT) can be performed, when using adaptations which minimize aerosolization, "based on individual institutional expertise and defined protocols". In practice, it is difficult to determine if tracheotomies have been a source of HCWs viral infection during the COVID-19 pandemics.

We thus hypothesized that both ST and PDT are safe for HCWs when procedures are adapted to minimize aerosol generation. If COVID-19-related adaptations were well designed, the two procedures should not lead to significant particle emission. To verify this hypothesis, this study aimed to monitor aerosol generation during ST and PDT, using COVID-19 adaptations to minimize aerosolization, in an acute respiratory distress syndrome (ARDS) swine model.

## 2 Materials and methods

### 2.1 Study design

In this physiological study, two randomized arms were compared (ST versus PDT) on a swine model to quantify aerosols generated during these procedures. Randomization was achieved before the beginning of the experiments, using the random function of Excel 2016 software (Microsoft, Redmond, Washington, USA). Each swine specimen served as its own control (baseline aerosol measurement).

### 2.2 Swine model

Institutional approval was obtained from the French ministry of higher education, research and innovation (n°APAFIS#26921–202008181721597, approval 2020–066). Twelve male Landrace pigs aged 3 to 5 months and weighing 55 kg to 90 kg underwent tracheostomy procedures (6 percutaneous and 6 surgical procedures). No exclusion criteria were set.

All pigs went from the same husbandry. They were housed in groups to limit their anxiety and stressed a week before procedures in same conditions with environmental enrichment (adapted toys like balls, biting ropes. . .). The pigs were fasted overnight, premedicated with intramuscular injection of ketamine (1.5 mg/kg) and midazolam (0.25 mg/kg) (Warner lambert, Nordic, AB Solna, Sweden) before transportation to the experiment facility. Sedation was deepened with propofol (2.5mg/kg, B Braun, Melsungen, Duitsland) via an ear vein cannula. After being placed in supine position, animals were intubated with a 7.5-mm internal diameter endotracheal tube (ETT) followed by injection of midazolam (0,3mg/kg) and cisatracurium (GlaxoSmithKline, Marly le Roy, France) infusion (bolus 0.5mg/kg). Then, pigs were connected to a ventilator with baseline settings set at a tidal volume (Vt) of 8 mL/kg, respiratory rate (RR) of 22 breaths/min, positive end-expiratory pressure (PEEP) of 5 cmH2O, and a fraction of inspired oxygen (FiO2) of 100%. The ventilator settings were then adjusted to the results of blood gas analyses performed along with the experiment. An initial rapid IV infusion of 1000 mL normal saline was given after anesthesia induction.

To simulate the worst (riskiest) conditions for HCWs, we maximized the probability of an aerosol generation by inducing an ARDS [30–32].

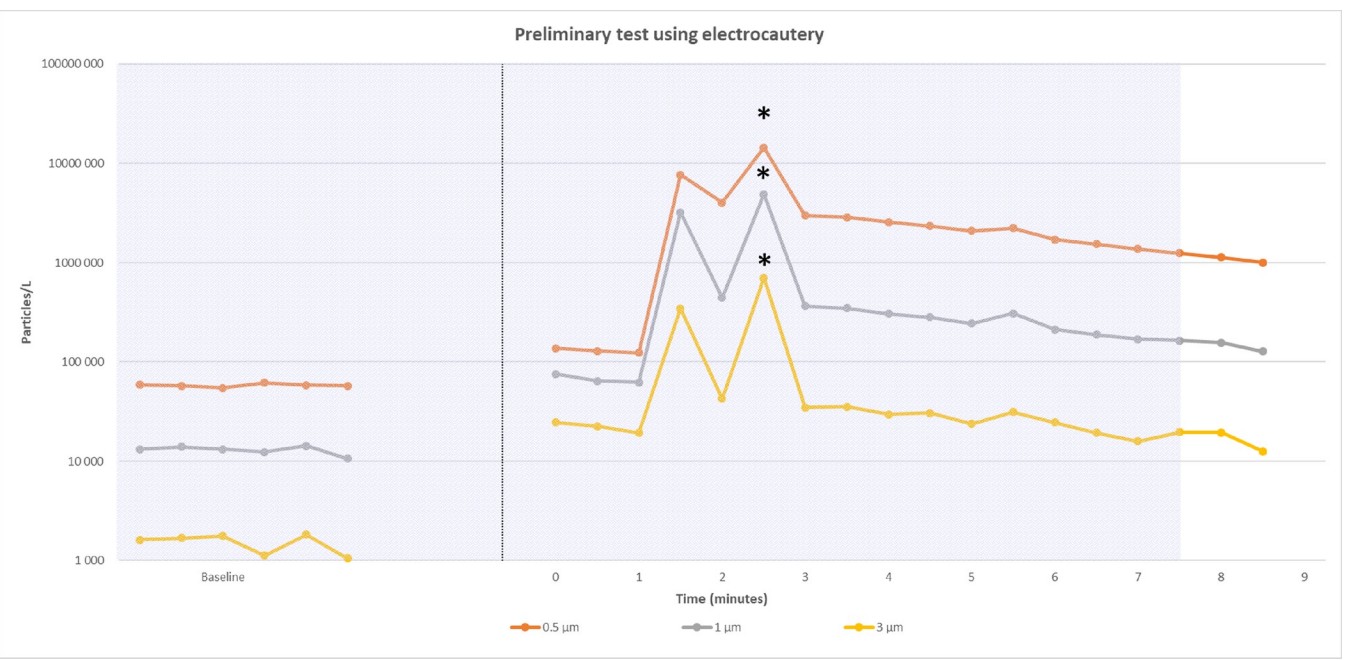

**Fig 1. Impact of electrocautery use on particle count during a preliminary test: Preliminary data experience showing the dramatic increase in particle count after electrocautery at T = 0.** Hatching in the background: Intensive care ventilator on; white in the background: Intensive care ventilator off; * significant peaks.

ARDS was performed as part of another research protocol (protocol available on request): this protocol provided ventilation conditions very similar to those of COVID-19 ARDS. Briefly, surfactant depletion was induced using repeated lung lavages (30 mL/kg warm 0.9% saline solution intratracheally) until PaO2/FiO2 was below 250 mmHg, followed by 2 hours of injurious ventilation (pressure controlled ventilation) with PEEP 0 cmH2O, inspiratory pressure 40 cmH2O, RR 10 bpm, and inspiratory:expiratory time ratio 1:1).

During preliminary tests, we observed that electrocautery could generate significant particle emissions, that could be mistaken for actual leaks and might bias the particle count (Fig 1), while the cardiac arrest did not modify the measured values. Therefore, cardiac arrest was induced before each procedure using potassium chloride to avoid bleeding. Time to desaturation (i.e., SpO2 <85%) and time to regain a correct saturation (i.e. >95%) were assessed after ARDS induction, prior to cardiac arrest, and were afterwards realistically simulated during the procedures.

## 2.3 Aerosol measurement

All procedures were performed in the same dedicated room to standardize aerosol measurements. The installation consisted in an operating table, an intensive care unit (ICU) ventilator (Dräger Evita Infinity V500, Lubeck, Germany), an optical particle counter (Met-One-3415, BeckMan Coulter, Brea, California, USA) generating a 28 L/min intake and a mobile air treatment unit (Fig 2).

The air exchange rate was 8.8 volumes per hour, ensuring decontamination kinetic (defined as the time required for the number of particles of 0.5–1 μm to be divided by 10) of 16 minutes (Fig 3). The particulate class defined by standard NF-EN-ISO-14644-1 according to the size of the particles present in the air was ISO 7, which meets ICU requirements and enables surgical procedures to be performed in the ICU [33]. Before each procedure, the room was

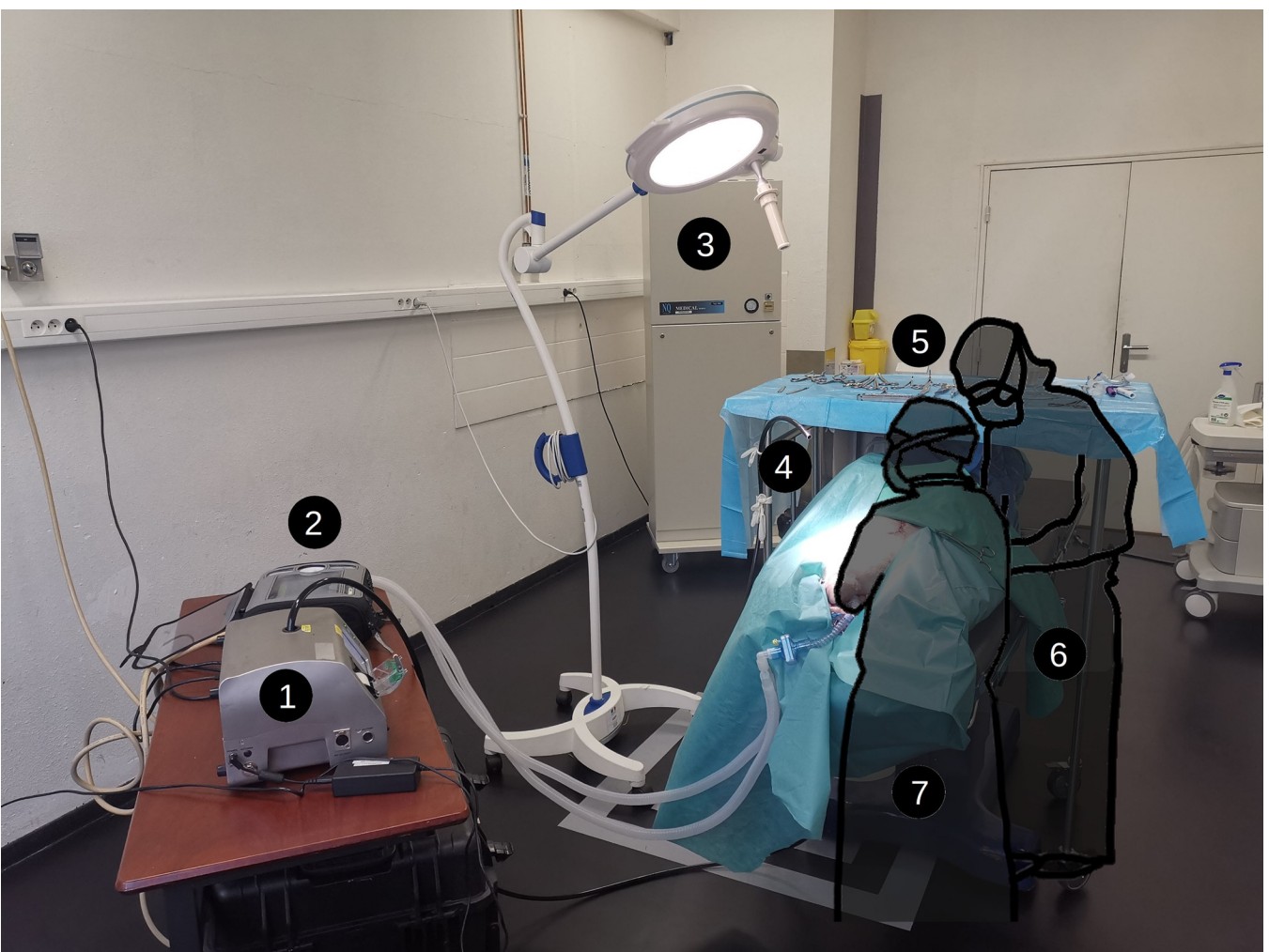

**Fig 2. Picture of the dedicated room and set-up for experiments 1: Optical particle counter; 2: Intensive care ventilator; 3: Mobile air treatment unit; 4: Optical particle counter sample pipe; 5: Instrument table; 6: First operator; 7: Assistant.**

decontaminated for a minimum of 50 minutes (equivalent to at least three decontamination kinetics). The particle counter collected (through a specific conductive sampling tube) different size rank of particles matter in situ (apparent diameter range 0.5–1 μm; 1–3; 3–5 μm, respectively designated as 0.5; 1; and 3 μm) emitted by the swine model. Particles < 1μm—of which viral particles—are expected to present the same behavior, a different behavior than those > 1μm due to Brownian motion.

The pipe was positioned between the tracheostomy area and the pig's head, near the operators' heads (i.e., 50 cm from the surgical site, see Fig 2). This area is presumably the most hazardous area for HCWs and where the maximum number of particles is likely to be generated (from either the endotracheal tube or the tracheostomy site). Then, the operators had to wait for a decontamination period (16 minutes) and a baseline measurement (at least 10 minutes) was performed to assess the initial level of particulate contamination due to the environment (dust convection, etc.), the monitored swine model and operators (which waited in the room).

During procedures, the particle contamination was sampled every 30 seconds (14 L) to evaluate the variations of particulate matter qualitatively and quantitatively. The continuous

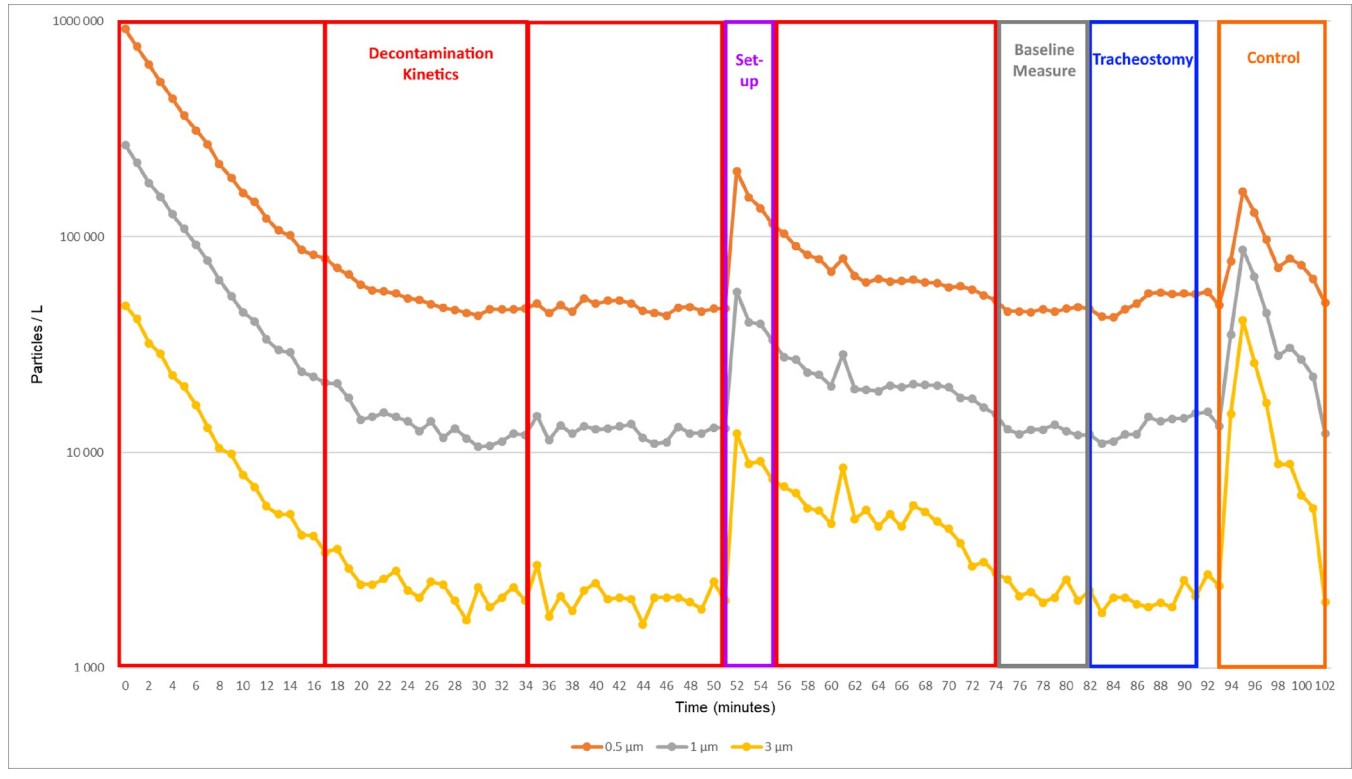

**Fig 3. Particle count variations (logarithmic scale) during decontamination kinetics, set-up, baseline measurement, tracheostomy procedure and control maneuver.** The sequence depicted here is as short as possible. At least three decontamination periods of 16 minutes before the set-up of the experiment were required. The set-up is generating airborne particles due to the entrance of the operators in the room. Thus, another decontamination period was performed before baseline measurement. 30 minutes after the set-up (equivalent to 2 decontamination kinetics), the operators were allowed to begin the procedure. An intentional aerosol-generating maneuver was then performed to control the effectiveness of airborne particle measurement.

measurement showed some variations of particle levels even after decontamination was performed, while no aerosol-generating maneuvers are performed. These baseline variations were assessed for all preliminary tests and tracheostomy experiments, to estimate the range of normal variations, due, for example, to the surgeon's moves in the room. Baseline measurements allowed us to interpret the events occurring during the experiments. Thus, an emission of particles was considered significant if the particle level increased beyond this level, i.e., by more than 10% from the baseline for 0.5 and 1 μm particles; and more than 25% for 3 μm particles (S1 Fig), meaning that the values correspond to emission of particles from the surgical field.

After each tracheostomy, an intentional aerosol-generating maneuver [34] was performed to ensure that aerosols were properly detected (control). It consisted of an intra-tracheal suctioning through the tracheostomy tube while mechanical ventilation was running. A comprehensive illustration of particle measurements for all steps is available in Fig 3.

## 2.4 Tracheostomy procedures

All procedures were fully video-recorded to analyze the timeline and match surgical steps to aerosol measurements data. All procedures were performed by expert ENT surgeons with experience well beyond the learning curve [35]. Two operators, with appropriate protective equipment, were required for each procedure, as well as an operator (VF) trained by ICU staff (ML and BP) to manage mechanical ventilation.

According to guidelines, respiratory settings during the tracheostomy procedure were performed to target protective ventilation as follows: FiO2 21%, Vt 6 mL/kg, RR 25/min and inspiratory:expiratory time ratio 1:2. PEEP was adjusted on i/plateau and ii/driving pressure [36]. When circuit disconnection was required, manual expiratory pauses were performed to prevent from excessive aerosol generation [23]. The expiratory pauses were used each time a potential leak in the ventilation circuit had to be managed. It corresponds to steps described for COVID-19 tracheostomy [3,16,23]: for each insertion of the fiber-optic endoscope in the endotracheal tube; puncture and dilatation steps; cannulation; circuit disconnection. The expiratory pauses were maintained for a maximal duration of 1 minute [23], and repeated as needed, in order to mimic clinical practice under hypoxic conditions. If the step was not achieved in a minute, the procedure was paused and the tracheostomy site was sealed to allow ventilation to be restored [37]. After each expiratory pause, a minimum of 1-minute ventilation was performed as recommended preventing desaturation [3]. For both ST and PDT, thyroid notch, sternal notch, cricoid cartilage and the first and second tracheal rings were marked on the skin with a surgical pen. Cuffed, non-fenestrated tracheostomy tubes were used as recommended [29]. The tracheostomy tube placement was confirmed with end-tidal CO2 and the circuit was checked for leaks before gently removing the endotracheal tube (ETT).

## 2.5 Open surgical tracheostomy (ST)

A modified ST minimizing the neck incision was performed, consisting of a 2.5-cm vertical incision from the level of the cricoid cartilage. Dissection proceeded through the platysma until the midline raphe between strap muscles. Strap muscles were separated and retracted laterally exposing the thyroid gland pushed inferiorly to allow a good exposure of the cricoid and first tracheal rings. An expiratory pause was performed during the following steps: advancement of the endotracheal tube, tracheal incision (inverted U-shaped opening), gently withdrawal of the ETT, tracheostomy tube insertion, cuff inflation, and connection to the ventilator circuit. Attention was paid to minimize suction steps during the entire procedure.

## 2.6 Percutaneous dilatational tracheostomy (PDT)

PDT procedures were performed according to the Ciaglia technique [38], modified for COVID-19 application [16,23,37]. A 1.5-cm vertical incision was performed at the level of the second tracheal ring. A slight dissection, using finger and Kelly clamp, was performed above the thyroid gland towards the trachea. A flexible endoscope was introduced in the endotracheal tube. The tube was gently pulled out until the inferior edge of the cricoid cartilage. The puncture through the neck incision was performed under endoscopic control, with a syringe half-filled with saline serum. The catheter was advanced while continuously applying negative pressure on the syringe until air bubbles are seen, confirming intratracheal placement. Then the syringe and needle were removed, catheter in place. A digital occlusion of the catheter was ensured to minimize leaks if ventilation needed to be restored. The guide wire was inserted in the catheter, and pre-dilator then dilator were used with gentle to-and-fro movement to achieve good dilatation. The tracheostomy tube was introduced under endoscopic guidance. The introducer was then replaced by the sleeve connected to the filter, the cuff was inflated, and the tracheostomy tube connected to the circuit.

## 2.7 Statistical analysis

Primary outcome was the occurrence of significant aerosol emission (leak event) during the procedure steps.

Secondary outcomes were the number of leak events, the duration of the procedure, the number of expiratory pauses, and technical problems (adverse events and/or human factors). Each significant event during procedures was interpreted in the light of the video data to correlate the emission with procedure steps and/or potential surgical mistakes. Student t-test was used to compare quantitative data, which were expressed in means +/- standard deviation.

Statistical analyses were carried out only at the completion of all procedures, using SPSS 24.0 for Windows 10.

## 3 Results

Twelve procedures were performed and registered (6 PDT and 6 ST). Two procedures (1 ST and 1 PDT) were excluded from the analysis due to artifacts and uncontrolled baseline variations presumably related to a lack of airtightness of the experimental room (window slightly ajar), leading to non-interpretable measurements (S1 Fig).

Particle count variations of the 10 analyzable tracheostomy procedures (named PDT-1 to PDT-5 and ST-1 to ST-5) are presented in Figs 4, 5 and S2, in which significant particle emissions are highlighted. For each procedure, the control event (intentional aerosol-generating suction) was positive. Finally, significant leaks were observed in 4/5 PDT (1 peak in PDT-1; 2 peaks in PDT-2; 1 peak in PDT-3; 3 peaks in PDT-4; no peaks in PDT-5) and 1/5 ST (1 peak in ST-3). However, the mean peaks of emitted particles were lower than those observed during provoked leaks (p<0.01) using an intra-tracheal suctioning (control, see Fig 6). We only observe for ST-3 a peak slightly higher than the control leak.

Mean procedure duration from incision to cannulation was 8.6 +/- 1.3 minutes for ST and 15.6 +/- 1.9 minutes for PDT (p = 0.0003). On average, PDT required 6.8 +/- 1.2 expiratory pauses versus 1 for ST (p<0.0001). The cuff was accidentally punctured (adverse event) during the first attempt of tracheal puncture in PDT-3 and led to significant particle peaks during the subsequent ventilation steps. A similar event occurred for ST-3 when entering the trachea, thus leading to a unique significant peak at the end of the procedure. For PDT-1, there was a delay in trans-tracheal catheter occlusion after the end of the expiratory pause (human error), responsible for a significant emission of particles. For PDT-2, the cannulation was associated with significant emission of particles from 0.5 μm to 3 μm. In several surgical procedures, the analysis of video data revealed the use of dry gauze pads simultaneously to the detection of particle increase. Small white dust particles coming from gauze pads are clearly visible on images may explain the rise in particle count (false positive, see S3 Fig). There was no aerosol detected after cannula fixation: thus, there were no leaks between the cannula and surrounding soft tissues in any experiment.

## 4 Discussion

In the context of COVID-19 outbreak, patients to HCW transmission of COVID-19 infection remained infrequent, probably due to the many procedure adaptations proposed to reduce the risk of aerosolization for both ST [11,12,20,21,39] and PDT [17,20,21,40,41] (Table 2). In particular, current guidelines aim to avoid respiratory circuit disconnection [17], and an end-expiratory pause should be performed to reduce aerosolization when a procedure step is likely to generate leaks [37]. Our study simulated procedures close to reality and demonstrates that performing both ST and PDT, with adaptations (using expiratory pauses according to guidelines in ICU settings) may not be so dangerous for HCWs, as stated by Sood et al. [42]. Indeed, there is no systematic aerosol leaks during the procedures; when present, these leaks are below or near the level of an intratracheal suctioning maneuver.

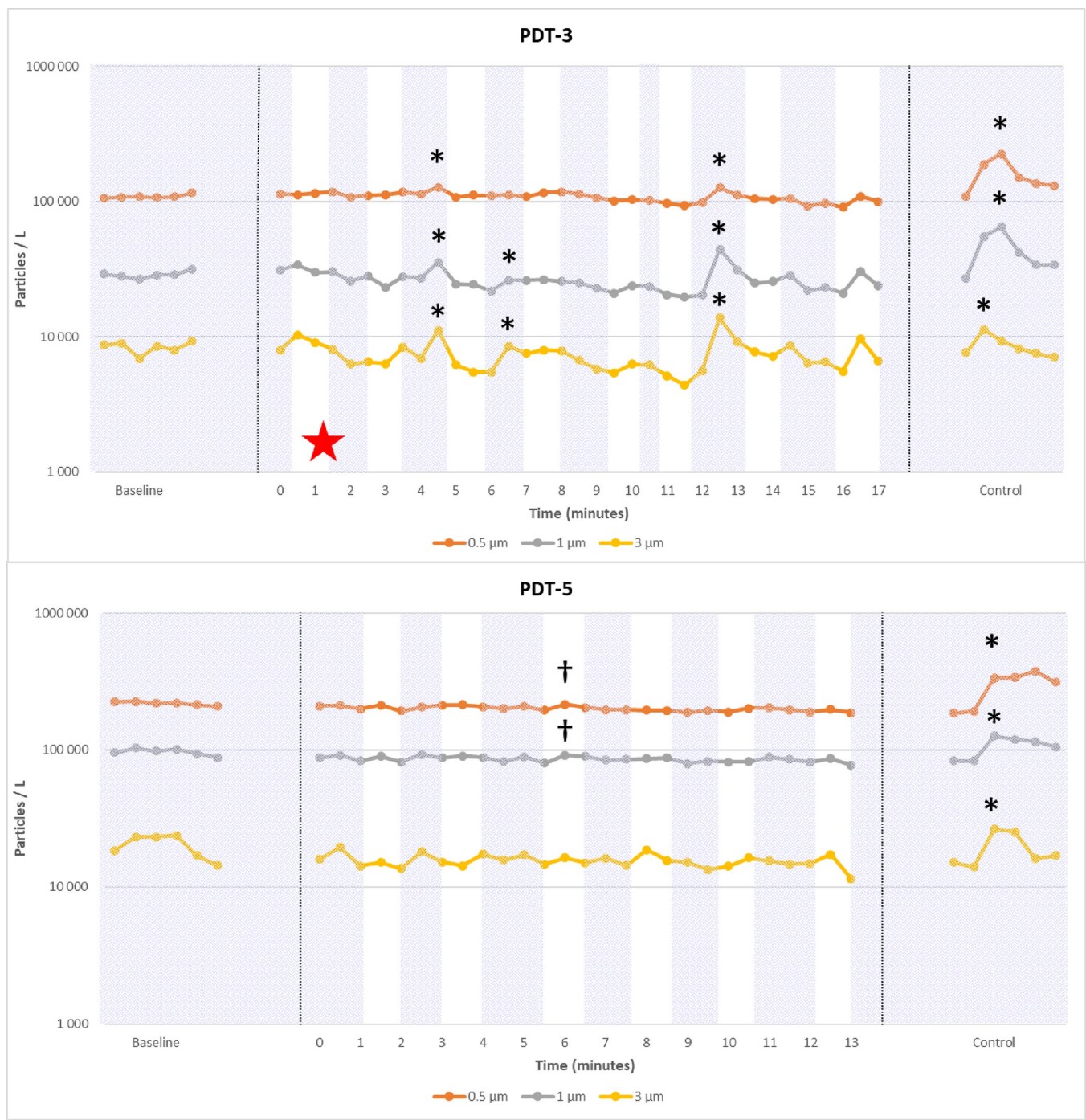

**Fig 4. Examples of particle count (logarithmic scale) during two percutaneous dilatational tracheostomy (PDT) procedures.** Hatching in the background: Intensive care ventilator on; white in the background: Intensive care ventilator off; * significant peaks related to a breach in ventilation circuit; † significant peaks related to an artifact (like dry gauze use); red star: Uneventful endotracheal tube cuff puncture. Baseline, procedure and intentional aerosol-generating maneuver (control) are shown. During PDT-3, an early cuff puncture was responsible of multiple leaks during the procedure, while no leaks occurred in PDT-5.

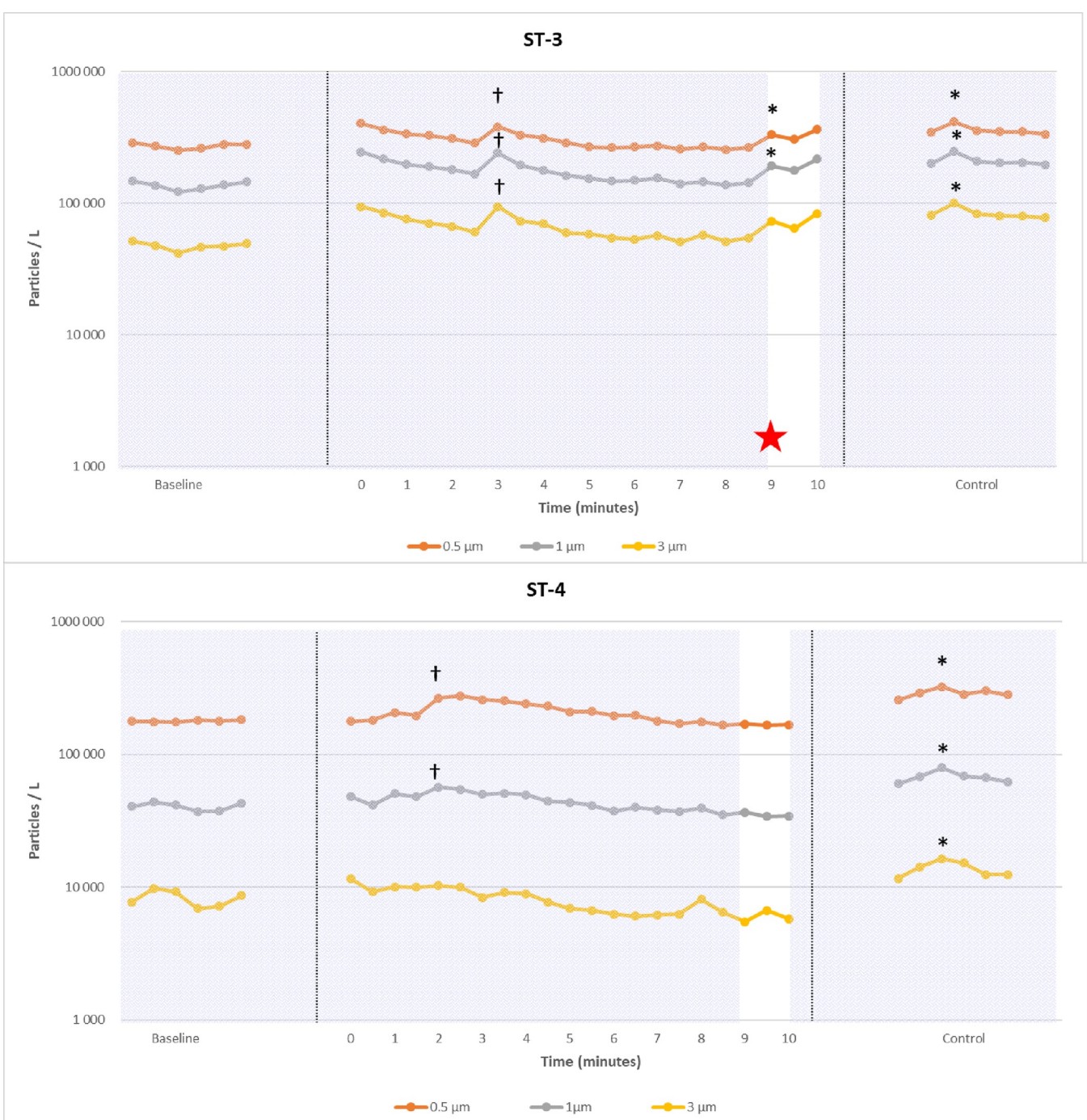

**Fig 5. Examples of particle count (logarithmic scale) during two surgical tracheostomy procedures (ST).** Hatching in the background: Intensive care ventilator on; white in the background: Intensive care ventilator off; * significant peaks related to a breach in ventilation circuit; † significant peaks related to an artifact (like dry gauze use); red star: Uneventful endotracheal tube cuff puncture. Baseline, procedure and intentional aerosol-generating maneuver (control) are shown. During ST-3, a late cuff puncture was responsible of 1 leak during the procedure, while no leaks occurred in ST-4.

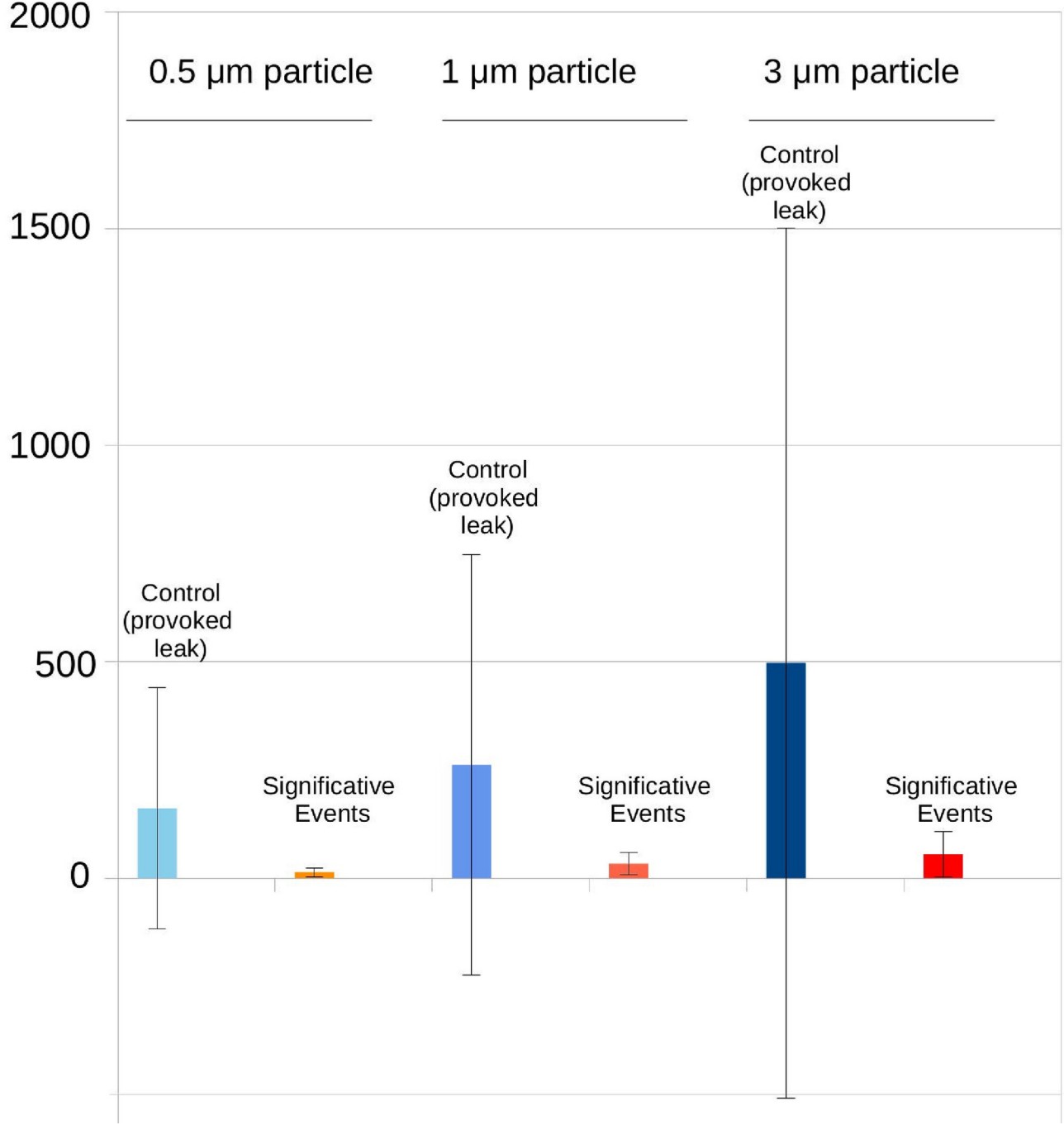

**Fig 6. Comparison of variations on particle count induced by control leaks and significant events during the whole procedures: The significant events generate lower levels of particles than controls.**

**Table 2. Reports of healthcare worker contamination during COVID-19 tracheostomy procedures in the literature (reports > 30 procedures).**

| Authors | Procedure type | Covid-19 adaptation | Number of procedures | Number of nosocomial contaminations reported |
|---|---|---|---|---|
| Picetti et al. [11] | ST | Yes | 66 | 0 |
| Avilés-Jurado et al. [12] | ST | Yes | 50 | 0 |
| Angel et al. [17] | PDT | Yes | 98 | 0 |
| Krishnamoorthy et al. [20] | PDT + ST | Yes | 143 (85 PDT + 58 ST) | 0 |
| Kwak et al. [21] | PDT + ST | For PDT only | 148 (NS) | 1* |
| Tetaj et al. [40] | PDT | Yes | 133 | 0 |
| Yokokawa et al. [39] | ST | Yes | 35 | 0 |
| Moizo et al. [41] | PDT | Yes | 36 | 0 |

* One otolaryngologist involved in open tracheostomy procedures was diagnosed COVID-19 positive but the causal link was not clearly established: 5 healthcare workers of its department fell ill too but were never involved in performing tracheostomies. PDT: Percutaneous Dilatational Tracheostomy; ST: Surgical Tracheostomy; NS: Non specified.

It may be difficult to achieve a perfectly airtight procedure from the beginning to the end, even when the operator is experienced. Breaches in the circuit of ventilation may occur during the tracheal puncture, tracheal incision, or in the case of ETT cuff perforation. This underlines the importance of having a good particle clearance in the room where the procedure is carried out, whatever the procedure chosen. In a recent comparison of 124 PDT versus 77 ST in the COVID-19 context, Rovira et al. [43] found four adverse events with potential exposure of HCWs to aerosol emission during the procedures (1 cuff rupture, 2 misplacements, 1 loss of airways), all of them in the PDT group. Niroula et al. [44] reported one out of 28 patients with an occurrence of circuit disconnection during a suture placement, exposing HCWs to aerosols. During procedures, iatrogenic cuff perforation may lead to an open-air system and HCWs aerosol exposure, as seen here for 'PDT-3.' This risk is present for both PDT and ST, but at different steps. ETT cuff perforation might occur at the end of ST (i.e., tracheal incision), and at the beginning of PDT (i.e., tracheal puncture). As a consequence, PDT could expose HCWs to aerosol leaks during the entire procedure. Ex-vivo swine model has already been used for ST [45], PDT training [46] and aerosol quantification [47,48]. In the swine model, it is not possible to simulate all the anatomical difficulties that can be encountered during a tracheostomy and that may greatly affect the safety of the procedure. However, the swine model also offers anatomical features that may hinder the procedure and may favor adverse cuff perforation with 1/ a smaller subglottic space than humans (resulting in a more difficult positioning of the ETT before tracheal puncture) 2/ thicker neck soft tissues preventing the use of trans-illumination for deciding the ideal level of puncture; 3/ a longer neck which may hinder the optimal placement of the ETT cuff 4/ a large thymus. Difficult anatomic conditions may be present in

**Table 3. Reports of puncture of the tracheal tube cuff during percutaneous dilatational tracheostomy procedures in the literature.**

| Authors | Number of procedures | Use of US guidance | % of tracheal tube cuff puncture |
|---|---|---|---|
| Holdgaard et al. [51] | 30 | No | 17% |
| Ahmed et al. [52] | 117 | No | 2.5% |
| Fikkers et al. [53] | 60 | No | 13.3% * |
| Guinot et al. [54] | 50 | Yes | 12% |
| Pattnaik et al. [55] | 300 | No | 4% |
| Khan et al. [56] | 56 | No | 1.78% |

* This % encompassed difficult puncture and/or punctured endotracheal tube.

human in real-life condition: to minimize the associated risks, it has been proposed to perform PDT with ultrasound guidance. In our model, this might have helped to prevent such adverse event [49,50], but we were not able to study it. Ultrasound guidance should probably be further encouraged for PDT in COVID patients. Nonetheless, the literature shows that ETT cuff perforation during PDT is not a rare issue and may occur in 2–12% of procedures (Table 3) [51–56]. Moreover, D'ascanio et al. [10] found that the air exposure time (i.e., the time interval between deflation of the ETT cuff and connection of the cuffed tracheostomy cannula to the ventilator) was longer in PDT (21.8 ± 5.7s) than in ST (5.5 ± 1.4 s). Thus, the risk of leaks during tracheostomy seems to be slightly greater for PDT, as proposed by McGrath [28]. Operator experience is essential to avoid such adverse event [35]. As this issue is frequent at the beginning of the learning curve [16], it is recommended not to involve trainees with COVID-19 cases unnecessarily [57].

PDT procedures are usually reputed to be shorter than surgical procedures [18,58] but, as pointed out by Riestra-Ayora et al. [18] it should be emphasized that there is a major indication bias: PDT is often reserved for patients with favorable anatomical conditions. Furthermore, procedural modifications related to COVID-19 substantially lengthen the duration of PDT procedures. Botti et al make the same observation as us: in the Covid-19 context, PDT procedures were longer than surgical tracheostomies (10–20' vs 30–45' in their experience). Nevertheless, another contributor of the short ST duration in our study may be that, in porcine model, the thyroid gland is smaller than in humans, allowing an infra-isthmic approach which is faster than trans-isthmic approach.

It is valuable to notice that the senior ENT surgeons are used to perform both ST and PDT in our center. They have respectively 7 and 15 years of experience in PDT and trained anesthesiologists and ENT staff to both techniques [16]. However, before the outbreak, ST was routinely performed while PDT remained a marginal indication: this may have also slightly contributed to fasten surgical procedures.

Patient instability due to ARDS is also a significant potential problem during tracheostomies, which should make a short procedure preferable. PDT is usually considered as faster than ST [27], but COVID-19 adapted PDT requires several pauses leading to longer-lasting procedures [58], and repeated interruptions of ventilation leading to potential desaturations. The decrease of alveolar ventilation shortly drives to hypoxemia in ARDS patients, which require to pause the procedure to allow reoxygenation before to resume. In the context of ARDS, desaturation episodes require careful monitoring and should be limited to as few as possible. ST, as it requires a single pause, may reduce per-procedure hypoxemia over PDT. ARDS condition, which often occurs in COVID-19 intubated patients, was reproduced to maximize the risk of particle emission (S1 Fig) and to better emulate patient instability. We were not able to directly assess SpO2 levels during procedures in our pulseless swine model nor to perfectly reproduce the stress generated among the team by these desaturations. However, in our simulation center, desaturations were simulated as realistically as possible, based on data obtained after ARDS induction and before cardiac arrest. One important other limitation is the absence of bleeding, which may also have favored ST procedures and may explain that these procedures were further accelerated. It is noteworthy that sample size (10 procedures) was too small to account for the many inadvertent events that may contribute to aerosolization in a larger sample or result in unforeseen events. In any case, many PDT steps are at risk of airway leaks, requiring repeated end-expiratory occlusions and specific actions (e.g., catheter occlusion with a finger): thus, the risk of human error (omission, communication, or synchronization problems) is likely to be increased.

Another question raised is the risk of contamination related to electrocautery use. In our model, we chose to study particle emissions without the use of electrocautery. The huge

amount of particle generated by electrocautery may have hindered the detection of particles coming from the airways (Fig 1), presumably the most dangerous one. Therefore, we decided to remove this potential confounder. Yet electrocautery, mostly used for ST, is likely to induce the release of particles that might carry viruses [59]. In a swine model, with a methodology similar to ours, Berges et al. highlighted a dramatic increase in particle emissions after electrocautery (increase of about 20–40 times the baseline) [48] and this release of particles might carry viruses [59]. In a study carried on surgical smoke in laparoscopy, Bogani et al. were indeed able to detect Sars-CoV-2 RNA [60]. Yet, it is still unclear if these particles contain viable viral material and if it carries a specific risk [61]. There is no evidence of viral transmission to HCWs from surgical smoke in pandemics. In the absence of definitive evidence, we can only recommend caution in the use of electrocautery [62,63] as the contamination risk cannot be totally ruled out [64]. A recent study [65] had compared aerosol and droplet scattering during tracheal incision for PDT and ST procedures, with measurements made in the 5 s period before the tracheal incision and the 5 s period after the tracheal incision. As recommended, mechanical ventilation was stopped when incising the trachea. There were significant leaks during PDT but no leak during ST procedures.

Another limitation relies on the OPC itself as its resolution only enables detection of particles of 0.5 microns and bigger. Considering the size of SARS-CoV-2 –ranging from 0.07 μm to 0.09 μm–, the smallest aerosol particle containing SarsCov2 virus could be as small as a single virion and would be indeed much smaller than those detected by OPC. Thus, although not plausible, it remains possible that isolated leaks of smaller particles ($<0.5\mu m$) may have occurred during the procedures. Yet, it should be emphasized that contaminant aerosols generated coughing or sneezing mostly contain particles of size 1–100 microns and that the maximum SARS-CoV-2 concentrations were measured in aerosol samples with diameters of 0.25–0.5 μm and 0.5–1.0 μm [66]. It should also be remembered that the objective of the study was not to achieve a precise quantification of the number of particles emitted.

Finally, our results seem to slightly favor ST over PDT (fewer aerosol leaks and shorter procedures). However, we would like to emphasize that, when deciding for PDT or ST, the problem is not limited to the technical part of performing tracheostomy: the discussion must take into account all aspects of the problem. For instance, if ST requires transportation of the patient to an operating room, this will require several intentional disconnections from the ventilator system, and there is a risk of extubation or accidental disconnection during transport. Bedside tracheostomy in the ICU in well-ventilated negative-pressure rooms is probably to be favored if possible [67]. Some authors also advocate that the PDT technique could be more airtight during the first few days after the procedure. However, with the adaptation of ST technique, our model showed no particle emission once the cannula was fixed. Similarly, Rovira et al. [43] reported only one leak around the cannula in the ST group (1/77) versus 0 in the PDT group. Again, a good particle clearance in ICU rooms appears crucial. Most importantly, attention must be paid to the timing of tracheostomy, as the risk of transmission increases with viral load [68].

Our initial hypothesis, that the adapted procedures for covid-19 were without risk of aerosolization for healthcare workers, could not be verified. Both PDT and ST remain at risk even if the measured peaks seemed inferior or comparable to an intratracheal suctioning maneuver. It is therefore important to recall that appropriate personal protective equipment and tracheostomy timing are major factors that help to minimize the risk of nosocomial infection for healthcare workers. Nosocomial Sars-Cov-2 infections during tracheostomy procedures seem to occur rarely (Table 2) thanks to appropriate personal protective equipment (PPE)–even if there is a risk of contamination during doffing or protection removal [69]–and specific adaptations of the procedure. Our results suggest that when all steps of the procedure are well

controlled, tracheostomies can be performed with minimal risk. Combined with the use of PPE and appropriate room ventilation, there is no reason to avoid conducting such procedures. It is crucial that patients do not suffer from an excess of precaution.

The COVID-19 strain slightly released since the vaccination era. However, preventing HCW from COVID19 exposure remains crucial as: 1—It prevents from sickness related absence and the consequences of a reduction in paramedical/medical human resources; 2— The staff may be composed of frailty members that could developed ARDS even after vaccination; 3—The efficacy of vaccination might vary along the years. It is also valuable to highlight that these results could be translated to other viral respiratory tract infection.

According to these results, we may assume what Botti et al. stated "If expert ENT surgeons are available, open ST might be preferred, since PDT could result in longer apnea and exposure to generated aerosols. However, authors recommend considering either open ST or PDT at the discretion of the medical staff involved in the procedure, according to their personal experience" [70].

## Supporting information

**S1 Fig. Normal variations of the baseline and induced by provoked leaks.** All data from percutaneous dilatational (PDT) and surgical tracheostomy (ST) procedures are shown. Mean baseline variation depends on size particles. For 0.5 and 1 µm particles, normal baseline variations were in 10% range. For 3 µm particles, normal baseline variations were in 25% range. PDT-0 and ST-0 were preliminary measures performed without acute respiratory distress syndrome (ARDS) induction. The level of particles emitted was 1–20 times higher using ARDS, which justified the systematic induction of ARDS. ST-6 and PDT-6 were excluded because of important baseline variations due to a slightly ajar window in the experiment room.
(PDF)

**S2 Fig. Particle count (logarithmic scale) during remaining percutaneous dilatational (PDT) and surgical tracheostomy (ST) procedures.** Hatching in the background: Intensive care ventilator on; white in the background: Intensive care ventilator off; * significant peaks related to a breach in ventilation circuit; † significant peaks related to an artifact (like dry gauze use). Baseline, procedure and intentional aerosol-generating maneuver (control) are shown.
(PDF)

**S3 Fig. Particles emitted during the use of dry gauzes (red circles) responsible of false positive peaks, as seen on video recording.**
(TIF)

## Acknowledgments

The authors acknowledge Marion Bernard, Vanessa Marie, Frédérique Groubatch, Annabelle Truck and Brice Mourer from the School of surgery of Nancy-Lorraine for their technical support.

## Author Contributions

**Conceptualization:** Valentin Favier, Patrice Gallet.

**Data curation:** Valentin Favier, Mickael Lescroart, Benjamin Pequignot, Patrice Gallet.

**Formal analysis:** Valentin Favier.

**Funding acquisition:** Valentin Favier.

**Investigation:** Valentin Favier, Patrice Gallet.

**Methodology:** Valentin Favier, Léonie Grimmer, Arnaud Florentin, Patrice Gallet.

**Resources:** Mickael Lescroart, Benjamin Pequignot.

**Supervision:** Patrice Gallet.

**Validation:** Léonie Grimmer, Arnaud Florentin.

**Visualization:** Patrice Gallet.

**Writing – original draft:** Valentin Favier, Patrice Gallet.

**Writing – review & editing:** Valentin Favier, Mickael Lescroart, Benjamin Pequignot, Léonie Grimmer, Arnaud Florentin, Patrice Gallet.

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
