## [Decision Letter · Decision Letter 0]

8 Sep 2022

PONE-D-22-21253Measurement of airborne particle emission during surgical and percutaneous dilatational tracheostomy COVID-19 adapted procedures in a swine modelPLOS ONE

Dear Dr. Favier,

Thank you for submitting your manuscript to PLOS ONE. After careful consideration, we feel that it has merit but does not fully meet PLOS ONE’s publication criteria as it currently stands. Therefore, we invite you to submit a revised version of the manuscript that addresses the points raised during the review process.

ACADEMIC EDITOR:

Thank you for submitting this interesting manuscript.

COVID-19 adaptations  should  be described in details in the method sections

Resolution of picture should be improved.

We look forward to receiving your revised manuscript.

Kind regards,

Silvia Fiorelli

Academic Editor

PLOS ONE

Journal Requirements:

Reviewers' comments:

Reviewer's Responses to Questions

**Comments to the Author**

1. Is the manuscript technically sound, and do the data support the conclusions?

Reviewer #1: Yes

Reviewer #2: Yes

2. Has the statistical analysis been performed appropriately and rigorously? 

Reviewer #1: Yes

Reviewer #2: I Don't Know

3. Have the authors made all data underlying the findings in their manuscript fully available?

Reviewer #1: Yes

Reviewer #2: Yes

4. Is the manuscript presented in an intelligible fashion and written in standard English?

Reviewer #1: Yes

Reviewer #2: Yes

5. Review Comments to the Author

Reviewer #1: The authors present a manuscript that aims to quantify aerosol generation during percutaneous tracheostomy as well as surgical tracheostomy using an optical particle counter. This topic has become of significant interesting during the COVID-19 pandemic as aerosolization of SARS-CoV-2 is thought to put the surgeon at risk of contracting COVID-19. Overall, this is a well-designed study using a porcine model of ARDS to determine the level particle generating during 5 ST and PDTs.

1. At many institution in the U.S. PDT’s are conducted by interventional pulmonary while ST are performed by ENT’s. Do the “expert ENT surgeons” routinely perform tracheostomy using both techniques? Can “beyond the learning curve” be expanded on – with an average of 10 years of experience? Or with at least ~# PDT and ST…

2. The PDT time was that much higher than ST time – is this consistent with actual surgical procedures? Can operative logs be assessed to see what a routine time to complete these procedures is for these surgeons performing a PDT and a ST from historical patient surgical logs or anesthesia records?

3. The authors frequently note various adaptations for PDT and ST to reduce aerosol generation in light of COVID-19 (Ln57-58, 71, 75, 78, etc)– these adaptations should be explicitly stated in the introduction. To highlight them for the reader a table, figure or specific paragraph maybe helpful.

4. Limitations of OPC should be discussed. Aerosol particles can be much smaller than 0.5um threshold readily detected by OPC and may limit particles that are quantified.

5. Figures (specifically the graphs) that were downloaded with the manuscript appeared low quality. The resolution made it difficult to read.

Minor:

1. Abstract Introduction: consider keeping PDT and ST in the same order in every sentence.

2. Line 73 - HCW “contamination” - is a strange way to phrase this. Later in the paper this is referred to as “nosocomial infection for health care workers” and it seems more intuitive to what the authors are talking about.

3. Ln 53 – consider changing “intensivist” to the “medical field”

4. Ln 56 – “contamination” – consider changing to “nosocomial infection”

5. Line 59 – clarify “learning societies” - relevant societies?

6. Line 107 – Is there a reference available for this protocol yet?

7. Line 133 – “thanks to a specific sample pipe” should be rephrased.

8. Ln 182 – Why was a vertical incision used for the ST? Is this recommended for COVID or is it a porcine adaptation of a typical horizontal incision.

9. Line 252 – term contamination is again used is this referring to HCW being infected? Or just getting particle on their attire.

10. Line 318 – “evacuate” consider use of “remove”

Reviewer #2: The authors have evaluated airborne particle emission during surgical and percutaneous dilatational tracheostomy using COVID-19 adaptations. I appreciate the authors’ effort to write this manuscript, but I have some points:

1- I suggest that you revise the title. At first the reader may misinterpret that you have only evaluated two tracheostomy methods for airborne particle emission.

2- Does your study add any helpful results for using in the clinical setting in this stage of the pandemic? As you know, health care workers have less concern about infection, especially after vaccination. I suggest that you write about it in the abstract and in the manuscript.

3- Please replace your pictures with high resolution ones.

6. PLOS authors have the option to publish the peer review history of their article (what does this mean?). If published, this will include your full peer review and any attached files.

Reviewer #1: No

Reviewer #2: No

---

## [Author Response · Author response to Decision Letter 0]

5 Oct 2022

Academic Editor:

Thank you for submitting this interesting manuscript.

COVID-19 adaptations should be described in detail in the method sections

Resolution of picture should be improved

• Comment: We thank the academic editor for these queries. COVID-19 adaptations have been described in a new table (Table 1), but in the introduction section, as required by Reviewer #1. Picture resolution has been improved.

Reviewer 1:

The authors present a manuscript that aims to quantify aerosol generation during percutaneous tracheostomy as well as surgical tracheostomy using an optical particle counter. This topic has become of significant interesting during the COVID-19 pandemic as aerosolization of SARS-CoV-2 is thought to put the surgeon at risk of contracting COVID-19. Overall, this is a well-designed study using a porcine model of ARDS to determine the level particle generating during 5 ST and PDTs.

• Comment 1: At many institutions in the U.S. PDT’s are conducted by interventional pulmonary while ST are performed by ENT’s. Do the “expert ENT surgeons” routinely perform tracheostomy using both techniques? Can “beyond the learning curve” be expanded on – with an average of 10 years of experience? Or with at least ~# PDT and ST

Answer 1: We thank reviewer 1 for this comment and agree with this observation: this situation is not the most prevalent. In France, PDT’s are mostly conducted by interventional pulmonary and anesthesiologists. In our center, one ENT surgeon (>15y experience) was previously trained in another center to flexible bronchoscopy for routine screening for synchronous malignancies in patients with HNSCCs and learned and daily practiced the PDT technique in this center (>50 procedures during this initial period). Subsequently, this surgeon continued to perform both techniques for years (either in ICU or in OR; but as these procedures were performed in 7 different departments, it is difficult to provide accurate data). Moreover, he trained the whole ENT staff and anesthesiologists (biannual inter-university difficult airway curriculum) to the PDT technique during all this period. The other senior surgeon had 7 years of experience with PDT and was also an instructor for PDT training courses on several occasions. 

We agree that the learning curve of PDT is not clearly defined in the literature. Some authors mentioned 20 procedures [ref 35] (as well as [10.1016/j.accpm.2016.07.005] for ultrasound-guided PDT) or 30 [10.1097/01.MLG.0000163744.89688.E8] procedures to achieve low complication rates. Our opinion is that this cut-off is not really sufficient, but other authors failed to demonstrate any relationship between complication occurrence and physician experience [https://pubmed.ncbi.nlm.nih.gov/11206898/]. Competence is not assessed solely on the basis of the number of procedures performed, which is why we did not insist on this debatable point. Therefore, based on these data, we wrote that “All procedures were performed by expert ENT surgeons with experience well beyond the learning curve“ (l163-164). 

Still, the indications for performing PDT rather than ST in the ENT department remained minority until the Covid outbreak (<5% of procedures performed by our team): then, the lack of anesthetists and the initial recommendations of the French ENT society (advocating for PDT) have led ENT specialists to perform many PDTs as soon as anatomical conditions were suitable (one per day during the epidemic peaks). 

However, we agree with reviewer 1, while ENT surgeons in our center routinely perform both techniques, before the COVID-19 outbreak, they had a greater experience with the surgical technique, and this might have also influenced the procedure duration. 

This is now specified in the discussion: nevertheless, it is still clear for us that seeking for an improved protection against particle leaks lengthens the PDT procedure, especially with unstable respiratory conditions, and that this is the main reason for the difference in procedure durations (see comment#2).

To clarify this specific point, the discussion section has been revised as follows.

Revised Manuscript – Lines 304-307: “It is valuable to notice that the senior ENT surgeons are used to perform both ST and PDT in our center. They have respectively 7 and 15 years of experience in PDT and trained anesthesiologists and ENT staff to both techniques. However, before the outbreak, ST was routinely performed while PDT remained a marginal indication: this may have also slightly contributed to fasten surgical procedures.

• Comment 2: The PDT time was that much higher than ST time – is this consistent with actual surgical procedures? Can operative logs be assessed to see what a routine time to complete these procedures is for these surgeons performing a PDT and a ST from historical patient surgical logs or anesthesia records?

Answer 2: Thank you for your comment. 

We are aware that these data may seem at odds with some case series outside the context of the Covid19 outbreak. 

While ST may be very quick in case of emergency (ST may be performed in <2 min , which is impossible with PDT (even if this may lead to a higher rate of complications…)), routine tracheostomies in surgeries for HNSCC (comparable to usual PDT conditions: with patients already intubated, without Covid19 nor anatomical difficulties) are usually performed by an experienced ENT surgeons in 10-15 minutes (this is easily verified in the recording of the schedule of the different surgical steps in OR), which is close to that of PDT in good conditions and outside the COvid19 context. Therefore, in our opinion, outside of this context, there is no reason to favor one technique or the other solely based on operative time (If one were to seek maximum efficiency, maybe we could even advocate for PDT if surgical tracheotomies would require a transfer to the operating room).

However, we agree with reviewer#1: several series seem to show that PDT are faster: it should though be emphasized that ST is usually preferred in case of anatomical difficulties: this is a bias which usually lengthens the procedure and which contributes to making PDT seem faster than ST. This bias has been pointed out in published series [for instance Riestra-Ayora et al.18]. This is also consistent with our usual clinical practice: the overwhelming majority of ST performed for ICU patients - in normal circumstances before the outbreak - are performed on patients with unfavorable anatomical conditions. The procedure can then be considerably lengthened. This is even more the case in an academic center, where many surgical tracheotomies are performed by junior surgeons under the supervision of seniors. In the porcine model, the thyroid gland is smaller than in humans, and surgical procedures were all performed using an infra isthmic approach: it may also have helped us to achieve faster operating times.

Yet, in our experience, the main explanation of the difference between ST and PDT lies in COVID19 adaptations: to seek for a maximal protection, PDT required several expiratory pauses and subsequent ventilation compensations to prevent desaturation3 while ST required only one pause. This may have led to an increase in the duration of PDT [10]. This is consistent with our clinical experience: we did not recorded the duration of all PDT procedures performed in the ICU during the outbreak but it was clearly lengthened (PDT procedures lasted approximately 30’ in average). Similar information are provided by Botti et al. [DOI: 10.1016/j.anl.2020.10.014 ] whom we quote here: “In our experience, operation time for OST ranges from 10 to 20′, while operation time for PDT is longer, ranging from 30 to 45′. Moreover, many severe COVID-19 patients needing invasive mechanical ventilation are obese and percutaneous tracheotomy could be actually challenging to perform in these patients.”

To clarify these points, the discussion section has been revised as follows:

Revised manuscript (lines 296-303): PDT procedures are usually reputed to be shorter than surgical procedures18,58 but, as pointed out by Riestra-Ayora et al.18 it should be emphasized that there is a major indication bias: PDT is often reserved for patients with favorable anatomical conditions. Furthermore, procedural modifications related to COVID-19 substantially lengthen the duration of PDT procedures. Botti et al make the same observation as us: in the Covid-19 context, PDT procedures were longer than surgical tracheostomies (10-20’ vs 30-45’ in their experience). Nevertheless, another contributor of the short ST duration in our study may be that, in porcine model, the thyroid gland is smaller than in humans, allowing an infra-isthmic approach which is faster than trans-isthmic approach. 

• Comment 3: The authors frequently note various adaptations for PDT and ST to reduce aerosol generation in light of COVID-19 (Ln57-58, 71, 75, 78, etc)– these adaptations should be explicitly stated in the introduction. To highlight them for the reader a table, figure or specific paragraph maybe helpful. 

Answer 3: We added these adaptations in a new table (Table 1).

Revised Manuscript – Line 656 : Table 1

• Comment 4: The Limitations of OPC should be discussed. Aerosol particles can be much smaller than 0.5um threshold readily detected by OPC and may limit particles that are quantified.

Answer 4: We thank the reviewer for this comment. Considering the size of SARS-CoV-2 - ranging from 0.07 μm to 0.09 μm -, the smallest aerosol particle containing SarsCov2 virus could be as small as a single virion and would be indeed much smaller than those detected by OPC. Yet in a study by Liu et al. (Nature. 2020;582:557–560. doi: 10.1038/s41586-020-2271-3), the maximum SARS-CoV-2 concentrations were measured in aerosol samples with diameters of 0.25–0.5 μm and 0.5–1.0 μm (the latter being detected by OPC). It should be remembered that the objective of the study was not to achieve a precise quantification of the number of particles emitted as a function of their size, but to detect potential leakage occurring during the procedure and to verify that procedural adaptations were able to limit these leakage moments. We were able to observe that the leaks were well detected, and that the quantities detected evolved in a joint and proportional way for all the sizes of particles detectable by the OPC. It seems to us implausible that leaks involving only particles smaller than 0.5 microns could have occurred during the procedures without any leakage of detectable particles larger than 0.5 microns: the majority of data suggests that size range of particles generated by coughing and sneezing by infected humans is from 1 µm to 100 µm. (Zhao et al., 2005; Han et al., 2013).Zhao, B., Zhang, Z., Li, X.T. (2005). Numerical study of the transport of droplets or particles generated by respiratory system indoors. Build. Environ. 40, 1032–1039. https://doi.org/10.1016/j.buildenv.2004.09.018 Han, Z.Y., Weng, W.G., Huang, Q.Y. (2013). Characterizations of particle size distribution of the droplets exhaled by sneeze. J. R. Soc. Interface 10, 20130560. https://doi.org/10.1098/rsif.2013.0560

However, we cannot completely rule out such a possibility: a specific paragraph was added in the text as follows: 

Revised Manuscript – Line 346-354: Another limitation relies on the OPC itself as its resolution only enables detection of particles of 0.5 microns and bigger. Considering the size of SARS-CoV-2 - ranging from 0.07 μm to 0.09 μm -, the smallest aerosol particle containing SarsCov2 virus could be as small as a single virion and would be indeed much smaller than those detected by OPC. Thus, although not plausible, it remains possible that isolated leaks of smaller particles (<0.5μm) may have occurred during the procedures. Yet, it should be emphasized that contaminant aerosols generated coughing or sneezing mostly contain particles of size 1-100 microns and that the maximum SARS-CoV-2 concentrations were measured in aerosol samples with diameters of 0.25–0.5 μm and 0.5–1.0 μm66. It should also be remembered that the objective of the study was not to achieve a precise quantification of the number of particles emitted.

• Comment 5: Figures (specifically the graphs) that were downloaded with the manuscript appeared low quality. The resolution made it difficult to read.

Answer 5: We thank the reviewer for this comment. To address this problem (which mainly concerned supplementary figures 1 and 2, figures were splitted and enlarged. Please find new version of the pics in the attached files. 

Concerning minor revisions: 

• Comment 1: Abstract Introduction: consider keeping PDT and ST in the same order in every sentence: 

Answer 1: we thank the reviewer for the accurate reading. Abbreviations have been replaced in correct order as suggested. 

• Comment 2: Line 73 - HCW “contamination” - is a strange way to phrase this. Later in the paper this is referred to as “nosocomial infection for health care workers” and it seems more intuitive to what the authors are talking about. 

Answer 2: Correction has been made as follow to clarify : “In practice, it is difficult to determine if tracheotomies have been a source of HCWs viral infection during the COVID-19 pandemics.”

• Comment 3: Ln 53 – consider changing “intensivist” to the “medical field”.

Answer 3: that is true. Change is done. 

• Comment 4: Ln 56 – “contamination” – consider changing to “nosocomial infection”

Answer 4: We thank the reviewer for comment. If accepted, we would prefer the term of “viral infection” as nosocomial infection rather report to patients than HCW. The change has been made as follow: Tracheostomies, as well as endotracheal intubation, are classified as aerosol-generating procedures1-3, exposing healthcare workers (HCWs) to a risk of viral infection4

• Comment 5: Line 59 – clarify “learning societies” - relevant societies?

Answer 5: the term “Learning societies” refers to academic scientific/medical associations editing guidelines in the medical field. If the term is confusing, we propose to remove the term without changing the meaning of the sentence as follow: “Despite many recommendations quickly issued by the learning societies to ensure the safety of HCWs performing tracheostomies[…].”

• Comment 6: Line 107 – Is there a reference available for this protocol yet?

Answer 6: Dear reviewer, the aim was to induce an ARDS to improve the external validity. There is no evidence in the literature that ARDS should increase aerosol emissions during tracheostomies. The protocol is mentioned in the reference number 32 and has already been published by our teams recently. 

• Comment 7: Line 133 – “thanks to a specific sample pipe” should be rephrased.

Answer 7: the sentence was rephrase as follow : “The particle counter collected (through a specific conductive sampling tube) different size rank of particles matter in situ (apparent diameter range 0.5-1 µm; 1-3; 3-5 µm, respectively designated as 0.5; 1; and 3 µm) emitted by the swine model.”

• Comment 8: Ln 182 – Why was a vertical incision used for the ST? Is this recommended for COVID or is it a porcine adaptation of a typical horizontal incision.

Answer 8: Even if the horizontal incision is mainly performed by surgical teams, some techniques use a vertical 1.5-2cm incision to minimize the trachea exposure and the risk of aerosolization [10].

• Comment 9: Line 252 – term contamination is again used is this referring to HCW being infected? Or just getting particle on their attire.

Answer 9: Dear reviewer, the term specifically refers to infection transmitted by patients to HCW while providing cares. To prevent from any misunderstandings, the sentence was rephrased as follows: “patients to HCW transmission of COVID-19 infection remained infrequent”

• Comment 10: Line 318 – “evacuate” consider use of “remove”

Answer 10: that is true again. Change is done.

 

Reviewer 2:

The authors have evaluated airborne particle emission during surgical and percutaneous dilatational tracheostomy using COVID-19 adaptations. I appreciate the authors’ effort to write this manuscript, but I have some points:

• Comment 1: I suggest that you revise the title. At first the reader may misinterpret that you have only evaluated two tracheostomy methods for airborne particle emission.

Answer 1: the title has been revised as follows: Measurement of airborne particle emission during surgical and percutaneous dilatational tracheostomy COVID-19 adapted procedures in a swine model: experimental report and review of literature

• Comment 2: Does your study add any helpful results for using in the clinical setting in this stage of the pandemic? As you know, health care workers have less concern about infection, especially after vaccination. I suggest that you write about it in the abstract and in the manuscript

Answer 2: Dear reviewer, thank you for your comment. That is right that the COVID-19 strain slightly released since the vaccination era. However, preventing HCW from COVID19 exposure remains crucial as: 

- It prevents from sickness related absence and the consequences of a reduction in paramedical/medical human resources

- The staff may be composed of frailty members that could developed ARDS even after vaccination

- This should be considered as “one more step” to prevent from COVID19 spreading in the hospital

- The efficacy of vaccination might vary along the years. 

Finally, it is valuable to highlight that these results could be translated to other viral respiratory tract infection. 

To address this specific comment, the discussion was adjusted as follows line 382-387: “the COVID-19 strain slightly released since the vaccination era. However, preventing HCW from COVID19 exposure remains crucial as: 1 - It prevents from sickness related absence and the consequences of a reduction in paramedical/medical human resources; 2 - The staff may be composed of frailty members that could developed ARDS even after vaccination; 3 - The efficacy of vaccination might vary along the years. It is also valuable to highlight that these results could be translated to other viral respiratory tract infection.”

And in the abstract: “Albeit the COVID19 strain slightly released since the vaccination era, preventing HCW from COVID19 infection remains a major economical and medical concern” 

• Comment 3: Please replace your pictures with high resolution ones.

Answer 3: thank you for your report. Blurry pictures have been optimized.

---

## [Decision Letter · Decision Letter 1]

10 Nov 2022

Measurement of airborne particle emission during surgical and percutaneous dilatational tracheostomy COVID-19 adapted procedures in a swine model: experimental report and review of literature

PONE-D-22-21253R1

Dear Dr. Favier,

We’re pleased to inform you that your manuscript has been judged scientifically suitable for publication and will be formally accepted for publication once it meets all outstanding technical requirements.

Kind regards,

Silvia Fiorelli

Academic Editor

PLOS ONE

Additional Editor Comments (optional):

congratulations to the authors and thanks to the reviewers for the suggestions provided which really helped improve the quality of the manuscript

Reviewers' comments:

Reviewer's Responses to Questions

**Comments to the Author**

1. If the authors have adequately addressed your comments raised in a previous round of review and you feel that this manuscript is now acceptable for publication, you may indicate that here to bypass the “Comments to the Author” section, enter your conflict of interest statement in the “Confidential to Editor” section, and submit your "Accept" recommendation.

Reviewer #1: All comments have been addressed

2. Is the manuscript technically sound, and do the data support the conclusions?

Reviewer #1: Yes

3. Has the statistical analysis been performed appropriately and rigorously? 

Reviewer #1: Yes

4. Have the authors made all data underlying the findings in their manuscript fully available?

Reviewer #1: Yes

5. Is the manuscript presented in an intelligible fashion and written in standard English?

Reviewer #1: Yes

6. Review Comments to the Author

Reviewer #1: Patient's adequately addressed all comments. This will be a nice contribution to the literature and of interest for ENT's, interventional pulmonologists, and intensivists.

7. PLOS authors have the option to publish the peer review history of their article (what does this mean?). If published, this will include your full peer review and any attached files.

Reviewer #1: No

---

## [Editor Report · Acceptance letter]

14 Nov 2022

PONE-D-22-21253R1 

Measurement of airborne particle emission during surgical and percutaneous dilatational tracheostomy COVID-19 adapted procedures in a swine model: experimental report and review of literature 

Dear Dr. Favier:

I'm pleased to inform you that your manuscript has been deemed suitable for publication in PLOS ONE. Congratulations! Your manuscript is now with our production department. 

Kind regards, 

on behalf of

Dr. Silvia Fiorelli 

Academic Editor

PLOS ONE